# Proteomic analysis of the umbilical cord in fetal growth restriction and preeclampsia

**Matthew S. Conrad**[1], **Miranda L. Gardner**[2], **Christine Miguel**[1], **Michael A. Freitas**[2], **Kara M. Rood**[1]*, **Marwan Ma'ayeh**[1]

**1** Department of Obstetrics and Gynecology, The Ohio State University College of Medicine, Columbus, Ohio, United States of America, **2** Campus Chemical Instrument Center (CCIC), Mass Spectrometry and Proteomics Facility, The Ohio State University College of Medicine, Columbus, Ohio, United States of America

* kara.rood@osumc.edu

**Data Availability Statement:** Proteomics data has been deposited to ProteomeXchange through massIVE and can be accessed through the dataset

## Abstract

Fetal growth restriction (FGR) is associated with adverse perinatal outcomes. Pre-eclampsia (PreE) increases the associated perinatal morbidity and mortality. The structure of the umbilical cord in the setting of FGR and PreE is understudied. This study aimed to examine changes in the umbilical cord (UC) composition in pregnancies complicated by FGR and FGR with PreE. UC from gestational age-matched pregnancies with isolated FGR (n = 5), FGR+PreE (n = 5) and controls (n = 5) were collected, and a portion of the UC was processed for histologic and proteomic analysis. Manual segmentation analysis was performed to measure cross-section analysis of umbilical cord regions. Wharton's Jelly samples were analyzed on a tims-TOF Pro. Spectral count and ion abundance data were analyzed, creating an intersection dataset from multiple mass spectrometry search and inference engines. UCs from FGR and FGR with PreE had lower cross-sectional area and Wharton's Jelly area compared with control (p = 0.03). When comparing FGR to control, 28 proteins were significantly different in abundance analysis and 34 in spectral count analysis (p < 0.05). Differential expression analysis between PreE with FGR vs controls demonstrated that 48 proteins were significantly different in abundance and 5 in spectral count. The majority of changes occurred in proteins associated with extracellular matrix, cellular process, inflammatory, and angiogenesis pathways. The structure and composition of the UC is altered in pregnancies with FGR and FGR with PreE. Future work in validating these proteomic differences will enable identification of therapeutic targets for FGR and FGR with PreE.

## Introduction

Fetal growth restriction (FGR) is a common condition that leads to a variety of adverse perinatal and postnatal outcomes. FGR is defined as an estimated fetal weight of less than the 10th percentile [1]. FGR increases the risk of neonatal morbidity and mortality including stillbirth as well as long term morbidity such as cognitive delay, obesity and coronary heart disease [2, 3]. FGR can be the result of maternal, fetal, and/or placental mechanisms. Of these, the most common mechanism is placental or umbilical cord disorders [4].

identifier PXD024751 or ftp link (ftp://massive. ucsd.edu/MSV000087051/)).

**Funding:** This work was supported by by the CCTS and CTSA Grant number (UL1TR002733) to MSC. https://ccts.osu.edu/content/current-grant-opportunities The tims-ToF Pro was purchased using funds from NIH award S10 OD026945. The funders had no role in study design, data collection and analysis, decision to publish, or preparation of the manuscript.

**Competing interests:** The authors have declared that no competing interests exist.

Pre-eclampsia (PreE) is also a major risk factor for development of FGR. PreE affects 3–8% of all pregnancies and leads to increased morbidity and mortality for both the mother and the fetus [5]. In addition to increased risk of FGR, PreE increases risk of preterm delivery, placental abruption, neonatal respiratory distress syndrome, cerebral palsy and perinatal death [5]. The etiology of PreE is multifactorial, and the pathophysiology of its fetal growth restrictive effects is not fully understood.

The umbilical cord connects the developing fetus to the placenta. The umbilical cord is composed of two umbilical arteries, one umbilical vein, and an extracellular matrix (ECM) surrounding these structures called Wharton's Jelly. It is thought that changes in the umbilical cord, including fibrosis, may lead to stricture or hypercoiling of the umbilical cord, obstructing uteroplacental blood flow [6]. Research is limited, however, on morphology of the umbilical cord, how the composition of Wharton's Jelly changes in FGR and PreE, and its role in these disease processes.

Wharton's Jelly is highly abundant in collagen and glycosaminoglycan (GAG) proteins [7]. In PreE, there is a shift to an increase in sulphated GAGs from hyaluronic acid. This shift in equilibrium may represent a "premature aging" of the tissue with contribution to the development of FGR [8]. However, little is known about the changes in the composition of Wharton's Jelly in FGR fetuses and warrants further investigation.

Umbilical cords from FGR pregnancies have demonstrated changes in the umbilical cord morphology including decreased umbilical cord diameters, cross-sectional area and decreased umbilical artery and vein area [9, 10]. These findings were further confirmed with ultrasound morphometry measurements [11]. Interestingly, the cross-sectional areas of the vessels are larger in the umbilical cords of babies born to women with PreE [12]. Yet, the volume of the whole umbilical cord and changes to the area of Wharton's Jelly in PreE has not been studied. These changes in morphology may contribute to the development of FGR due to changes in fetal blood flow.

Mass spectrometry-based proteomics has emerged as a promising high-throughput technology for identification of potential biomarker candidates for diseases. Recent reviews have highlighted the proteomic approaches that have been used to explore PreE, FGR and preterm birth [13–15]. This data has been complicated by a wide variety of techniques used and tissues analyzed. Despite this variability, meta-analysis has identified common proteins across multiple studies that are differentially expressed [14]. None of these studies, however, have looked at the Wharton's Jelly and the differential protein expression in FGR and PreE.

The study's primary hypothesis is that the umbilical cord structure and proteomic composition of Wharton's Jelly is altered in women with FGR and PreE + FGR. Specifically, these changes will include decreased cross-sectional area of the umbilical cord and proteomic changes in extracellular matrix proteins.

## Methods

### Screening and enrollment

Patients who presented to labor and delivery with singleton gestations were considered for this study. The study design was a case-control study with groups as follows: FGR, FGR with PreE, and gestational age-matched controls (n = 5 for each group). Gestational age was determined by last menstrual period and ultrasound biometry before 20 weeks using standard criteria [16]. Patients were consented for the study by trained research personnel and clinical care was at discretion of their provider. Demographic data was collected through a combination of interview with the patient and the electronic medical record. Delivery and neonatal outcomes were assessed through the electronic medical record. Data collected included gestational age,

maternal age, parity, delivery route, Body Mass Index, chronic hypertension, diabetes, neonatal APGAR scores, neonatal sex, and neonatal weight. Subjects were recruited, and written informed consent obtained, under the Ohio State University Institutional Review Board (Study Number: 2019H0039).

## Sample collection

The umbilical cords were collected after delivery and placed under refrigeration within one h. A standardized 1cm segment was collected approximately 5cm from the cord placental insertion site and placed in 10% formalin for histology within 6 h from delivery. Wharton's Jelly was dissected from the umbilical cord, immediately frozen at -80˚C and stored until further processing.

## Umbilical cord histological analysis

The 1cm segments of umbilical cord were previously placed in 10% formalin. The samples were processed by the Comparative Pathology and Mouse Phenotyping Shared Resource. They were embedded in paraffin, sectioned at 4μm, and then stained with hematoxylin and eosin. They were then digitally scanned by using an Aperio Digital Pathology System (Leica, Illinois, USA). ImageJ was used for area analysis of the umbilical cord using manual segmentation. Cross sectional area measurements were computed for umbilical cord area, Wharton's Jelly area, average artery tunica media outer, average artery tunica media inner, and vein wall area.

## Statistical analysis

Statistical analysis was performed using SPSS (IBM, Armonk, NY, USA). Baseline patient characteristics were compared using one-way ANOVA with Bonferroni post-hoc testing. The APGAR data is presented as medians with interquartile range with comparison using a Kruskal-Wallis test. Umbilical cord region of interest area means were compared with a general linear model with Bonferroni post-hoc testing with correction for gestational age. Significance threshold was considered at p-value < 0.05.

## Sample preparation for proteomics

At time of collection, a portion of Wharton's Jelly was dissected and frozen at -80˚C. Samples were washed twice with 50 mM ammonium bicarbonate. Approximately 43 mg of Biorupter sonication beads were then added to the sample, along with 100 μL of 50 mM ammonium bicarbonate containing 0.1% Rapigest (Waters Corp). Samples were sonicated in a Biorupter (Diagenode) with 20 on/off cycles using 30 sec on and 30 sec off. Extracts were spun at 13K rpm in a microcentrifuge to pellet debris and supernatant was transferred to a new tube. The supernatant was incubated with DTT (5 mM final concentration) at 65˚C for 30 minutes. The supernatant was then incubated with iodoacetamide (15 mM final concentration) in the dark at room temperature for 30 minutes. Trypsin (1 ug, sequencing grade, Promega) was added for digestion, and samples were then incubated at 37˚C overnight. The following day, digestion was quenched with addition of trifluoroacetic acid (final concentration 0.5%) and sample was incubated at 37˚C for 30 minutes to precipitate the Rapigest. The sample was clarified at 13K rpm for 5 min in a microcentrifuge, supernatant dried down in a vacufuge (Eppendorf), and desalted with a Ziptip. After desalting, samples were dried down in a vacufuge prior to resuspension in water with 0.1% formic acid and determination of peptide concentration via nanodrop (A280nm).

## LC-MS/MS

Protein identification was performed on the supernatant from the protein digestion of the Wharton Jelly samples using nano-liquid chromatography-nanospray tandem mass spectrometry (LC/MS/MS) on a Bruker tims-ToF Pro equipped with a CaptiveSpray source operated in positive ion mode. Samples (200ng injection) were separated on a $C_{18}$ reverse phase column (1.6 μm, 250mm* 75 μm IonOpticks) using a Bruker nanoElute UHPLC system. Pre-injection column equilibration consisted of 4 column volumes at 800 bar. Mobile phase A was 0.1% Formic Acid in water while acetonitrile (with 0.1% formic acid) was used as mobile phase B. A flow rate of 0.4 μL/min was used. Mobile phase B was increased from 2 to 17% over the first 60 min, increased to 25% over the next 10 min, further increased to 37% over the next 10 min, and finally increased to 80% over 10 min and then held at 80% for 10 min.

MS and MS/MS experiments were recorded over the *m/z* range 100–1700 and $K_0$ of 0.6–1.6. PASEF was used for all experiments, with the number of PASEF MS/MS scans set to 10. Active exclusion was applied, releasing after 0.4 min, with precursor reconsidered if current intensity/previous intensity was 4.0 or greater.

## Protein identification, data analysis and statistics for proteomics

For this study, a bottom-up shotgun quantitative proteomic approach was considered for identification of differentially expressed proteins in Wharton's Jelly. A combination of label-free quantification strategies, peak intensity and spectral count [17, 18], were used to analyze the data. Briefly, raw.d files generated from the timsTOF Pro were converted to mzML with OpenMS (v 2.5.0) and tdf2mzml in-house nextflow script. Converted mzML files were searched against a reviewed UniProt human proteome (downloaded 1/1/2020) in OpenMS with the following protein search engine and inference engine combination: Comet fido, Comet epiphany, X!Tandem epiphany, MSGF+ fido, and MSGF+ epiphany [19–22]. Search parameters included precursor mass tolerance 20 ppm, MS2 mass tolerance 0.05 Da, carbamidomethylation of cysteine as a fixed modification, oxidation of methionine as a variable modification and false discovery rate (FDR) of peptide and proteins equal to 0.05.

For differential expression analysis with spectral count data, samples were trimmed mean normalized, and significance (p-value < 0.05) determined by edgeR generalized linear model quasi-likelihood Ftest (glmQLFTest) as described in [23]. Resulting p-values were adjusted for multiple hypothesis testing with Bonferroni-Hochberg method. For differential expression analysis with peak intensity data, missing values were imputed according to sample group, described by Gardner *et al.* [24]. Data was quantile normalized, and significance (p-value < 0.05) determined by a modified exact test. Downstream gene ontology, KEGG and pathway analyses utilized the list of significant proteins identified across the database combination for each pair-wise comparison, reporting the log fold-change values as well. Protoemics data has been deposited to ProteomeXchange through massIVE and can be accessed through the dataset identifier PXD024751 or ftp link (ftp://massive.ucsd.edu/MSV000087051/).

## Results

### Histology

Characteristics of the study population are shown in Table 1. There was no significant difference found in the maternal characteristics including maternal age, BMI, presence of chronic hypertension, or maternal diabetes (pre-gestational or gestational). There was no significant difference in fetal gestational age at time of delivery (p = 0.36). There was a significant difference in route of delivery where all the controls were vaginal deliveries and the FGR and FGR

**Table 1. Demographics and clinic assessment of study population.**

|  | Mean (SD) Control | FGR | FGR + PreE | P-Value |
|---|---|---|---|---|
| Number | 5 | 5 | 5 | - |
| Gestational Age (wks) | 35.1 (3.3) | 35.3 (3.1) | 32.9 (1.4) | 0.362 |
| Maternal Age (years) | 26.2 (4.8) | 31.6 (5.0) | 29 (10.5) | 0.520 |
| Parity | 1.8 (0.4) | 1.4 (0.5) | 1.2 (0.4) | 0.397 |
| Delivery Route - Vaginal | 5 | 1 | 1 | **0.006** |
| - Cesarean | 0 | 4 | 4 | |
| BMI | 27.5 (6.5) | 36.2 (9.2) | 40.2 (12.9) | 0.211 |
| cHTN | 0 | 1 | 2 | 0.335 |
| Diabetes | 1 | 2 | 2 | 0.422 |
| Neonate Sex - Male | 4 | 4 | 1 | 0.088 |
| - Female | 1 | 1 | 4 | |
| Neonate Weight (g) | 2361.2 (550.6) | 1944.20 (810.8) | 1431.6 (351.40) | 0.088 |
|  | Median (25%ile,75%ile) | | | |
| AGARS 1 min | 8 (7,8) | 8 (6,9) | 8 (2,8) | 0.781 |
| APGARS 5 min | 9 (9,9) | 9 (8,9) | 9 (9,9) | >0.999 |

with PreE were majority cesarean delivery (p < 0.01). The indication for cesarean delivery was non-reassuring fetal testing in all cases. The indication for delivery for the controls were spontaneous preterm labor or PPROM. There were no significant differences in 1 min and 5 min APGAR scores, parity, or neonatal sex between groups. There was a trend for difference in neonatal weight with FGR and FGR with PreE less than controls (p = 0.09).

Histological analysis of the umbilical cord showed that the cross-sectional total umbilical cord area is significantly different between the groups with the control groups having a significantly larger area compared to the FGR and FGR+PreE groups (Fig 1, p = 0.03) when corrected for gestational age. Similarly, there is a significant difference in the Wharton's Jelly area with the control groups having a significantly larger area compared to the FGR and FGR+PreE groups (Fig 1, p = 0.03). There were no significant differences found in the artery average tunica media outer area (S1 Fig). The average of the FGR with PreE arterial tunica media inner

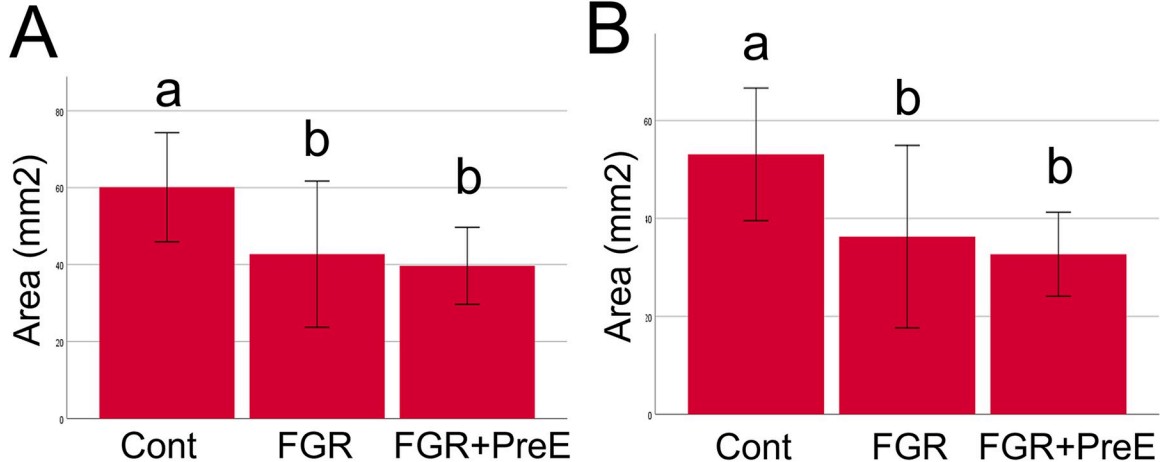

**Fig 1. Umbilical cord total area and Wharton's Jelly area.** Umbilical cord total area with groups control, FGR, FGR + PreE (A, p = 0.03). Wharton's Jelly area (B, p = 0.03). Fetal Growth Restriction (FGR), Preeclampsia (PreE). Bonferroni post-hoc analysis with significantly different groups indicated by letter above bar.

layer was larger than controls and FGR, but this did not reach significance (S2 Fig, p = 0.75). There were no differences in the umbilical cord vein wall area (S3 Fig).

## Proteomics

A novel comparison technique for identification of proteins was used in this study. Five combinations of protein identification databases (Comet fido, Comet epiphany, X!Tandem epiphany, MSGF+ fido, and MSGF+ epiphany) were used to generate a common overlap dataset containing over 1000 proteins present in each pair-wise comparison. As seen in Fig 2, peak intensity analysis for FGR versus controls resulted in 28 common differentially expressed proteins across the five different database combinations. Fourteen proteins were downregulated

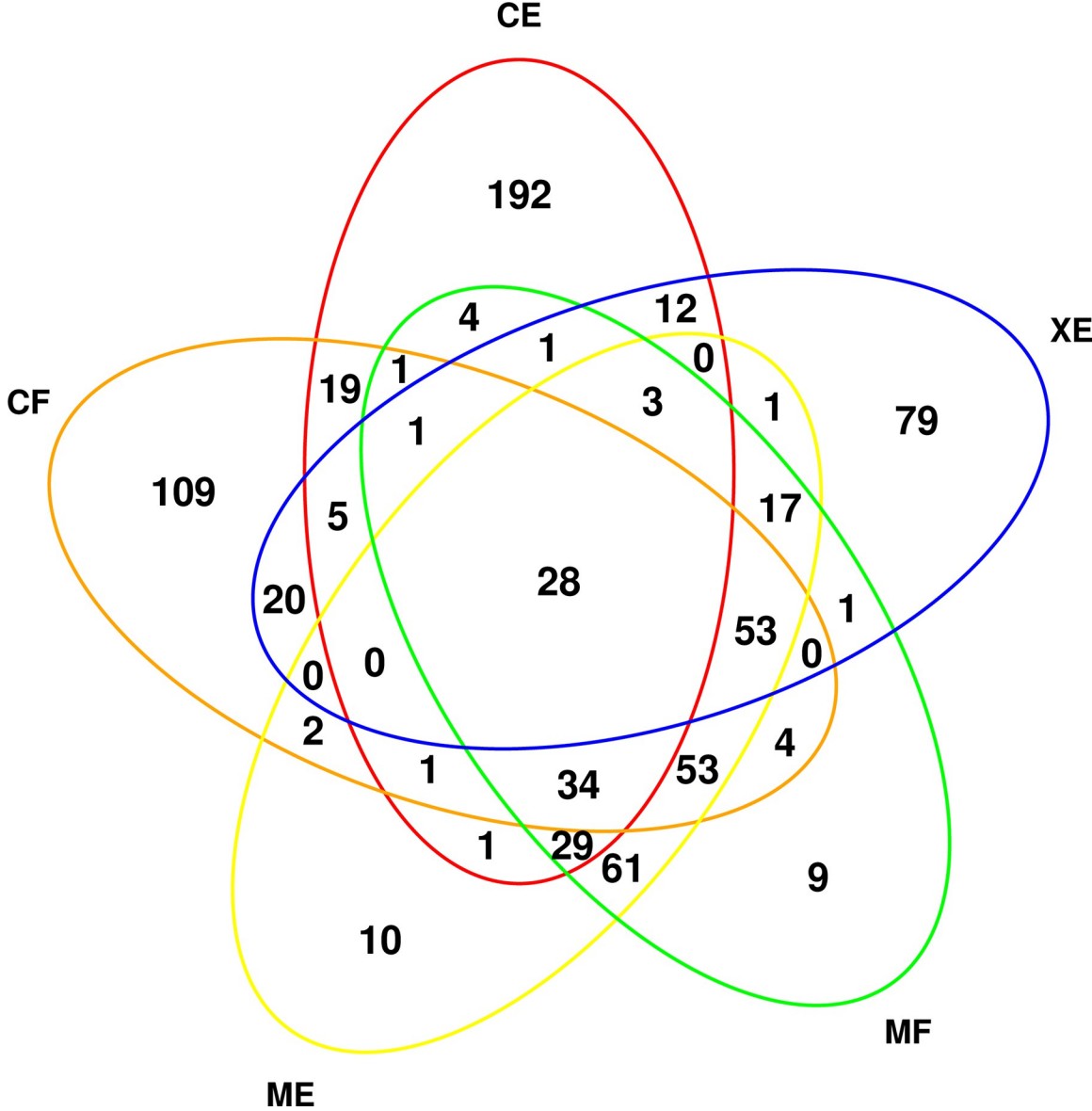

**Fig 2. Overlap of significantly changed protein ID's from mass spectrometry search and inference engines for peak intensity of FGR versus controls.** Legend (CF: comet fido; CE: comet epifany; XE: X!Tandem epifany; MF: MSGF fido; ME: MSGF epiphany).

**Table 2. Intersection of significantly changed protein expression for FGR versus controls spectral count.**

| UniProt ID | Protein | Log Fold Change |
|---|---|---|
| Q9NR99 | Matrix-remodeling-associated protein 5 | -3.32 |
| P20742 | Pregnancy zone protein | -3.13 |
| O75339 | Cartilage intermediate layer protein 1 | -2.90 |
| Q32P28 | Prolyl 3-hydroxylase 1 | -2.61 |
| P12107 | Collagen alpha-1(XI) chain | -2.49 |
| P25940 | Collagen alpha-3(V) chain | -2.39 |
| P49746 | Thrombospondin-3 | -2.16 |
| Q12884 | Prolyl endopeptidase FAP | -2.14 |
| P62263 | 40S ribosomal protein S14 | -2.06 |
| P78539 | Sushi repeat-containing protein SRPX | -2.00 |
| A1L4H1 | Soluble scavenger receptor | -1.88 |
| Q13421 | Mesothelin | -1.85 |
| P55287 | Cadherin-11 | -1.57 |
| Q14257 | Reticulocalbin-2 | -1.36 |
| P01861 | Immunoglobulin heavy constant gamma 4 | -1.27 |
| Q07092 | Collagen alpha-1(XVI) chain | -1.17 |
| P02771 | Alpha-fetoprotein | -0.98 |
| Q02809 | Procollagen-lysine,2-oxoglutarate 5-dioxygenase 1 | -0.96 |
| Q9NRN5 | Olfactomedin-like protein 3 | -0.93 |
| Q96CN7 | Isochorismatase domain-containing protein 1 | -0.91 |
| Q9HCB6 | Spondin-1 | -0.74 |
| O00339 | Matrilin-2 | -0.50 |
| P02452 | Collagen alpha-1(I) chain | -0.43 |
| Q14112 | Nidogen-2 | 1.06 |
| P00390 | Glutathione reductase, mitochondrial | 1.18 |
| P126112 | Aggrecan core protein | 1.26 |
| P31040 | Succinate dehydrogenase | 1.31 |
| Q7Z4W1 | L-xylulose reductase | 1.72 |
| O14672 | Disintegrin and metalloproteinase domain-containing protein 10 | 2.09 |
| P48960 | Adhesion G protein-coupled receptor E5 | 2.16 |
| Q9HAV0 | Guanine nucleotide-binding protein subunit beta-4 | 2.38 |
| P36269 | Glutathione hydrolase 5 proenzyme | 2.39 |
| Q8WUP2 | Filamin-binding LIM protein 1 | 2.43 |
| P42226 | Signal transducer and activator of transcription 6 | 3.00 |

and fourteen upregulated with the largest log fold-change (logFC) values include Prolyl 3-hydroxylase 3, Immunoglobulin heavy constant gamma 4, Procollagen C-endopeptidase enhancer 1, COP9 signalosome complex subunit 3 and Cyclin-dependent kinase 6 (S1 Table). Thirty-four proteins were significantly changed for FGR versus controls using spectral count. Twenty-three were downregulated and eleven upregulated with largest logFC included Matrix-remodeling-associated protein 5, Pregnancy zone protein, Signal transducer and activator of transcription 6, and Filamin-binding LIM protein 1 (S4 Fig, Table 2). Forty-eight proteins were significantly changed for FGR with PreE versus controls using peak intensity (S5 Fig) with twenty being downregulated and twenty-eight being upregulated. The proteins with the largest logFC included Eukaryotic translation initiation factor 3 subunit E, Protein enabled homolog, Desmocollin-2and Tripeptidyl-peptidase 2 (Table 3). Five proteins were significantly changed for FGR with PreE versus controls using spectral count with two being downregulated

**Table 3. Intersection of significantly changed protein expression for FGR+PreE versus controls peak intensity.**

| UniProt ID | Protein | Log Fold Change |
|---|---|---|
| Q8N8S7 | Protein enabled homolog | -4.07 |
| P60228 | Eukaryotic translation initiation factor 3 subunit E | -2.65 |
| P62191 | 26S proteasome regulatory subunit 4 | -2.54 |
| Q13576 | Ras GTPase-activating-like protein IQGAP2 | -2.26 |
| Q16881 | Thioredoxin reductase 1, cytoplasmic | -2.09 |
| Q02818 | Nucleobindin-1 | -1.93 |
| O95373 | Importin-7 | -1.90 |
| P23381 | Tryptophan—tRNA ligase, cytoplasmic | -1.90 |
| P56134 | ATP synthase subunit f, mitochondrial | -1.88 |
| Q9BTT0 | Acidic leucine-rich nuclear phosphoprotein 32 family member E | -1.84 |
| Q9Y446 | Plakophilin-3 | -1.82 |
| Q13361 | Microfibrillar-associated protein 5 | -1.77 |
| P30711 | Glutathione S-transferase theta-1 | -1.58 |
| P04181 | Ornithine aminotransferase, mitochondrial | -1.49 |
| P01721 | Immunoglobulin lambda variable 6–57 | -1.45 |
| P09110 | 3-ketoacyl-CoA thiolase, peroxisomal | -1.37 |
| P06276 | Cholinesterase | -1.36 |
| P13693 | Translationally-controlled tumor protein | -1.34 |
| P55001 | Microfibrillar-associated protein 2 | -1.26 |
| Q9Y3Z3 | Deoxynucleoside triphosphate triphosphohydrolase SAMHD1 | -1.12 |
| Q8NB37 | Glutamine amidotransferase-like class 1 domain | 1.09 |
| P27169 | Serum paraoxonase/arylesterase 1 | 1.13 |
| Q14192 | Four and a half LIM domains protein 2 | 1.39 |
| Q13325 | Interferon-induced protein with tetratricopeptide repeats 5 | 1.47 |
| Q96CX2 | BTB/POZ domain-containing protein KCTD12 | 1.47 |
| P35268 | 60S ribosomal protein L22 | 1.49 |
| P01834 | Immunoglobulin kappa constant | 1.49 |
| Q6XQN6 | Nicotinate phosphoribosyltransferase | 1.67 |
| O75489 | NADH dehydrogenase [ubiquinone] iron-sulfur protein 3 | 1.79 |
| P20810 | Calpastatin | 1.84 |
| P52566 | Rho GDP-dissociation inhibitor 2 | 1.88 |
| Q8TDZ2 | [F-actin]-monooxygenase MICAL1 | 1.94 |
| P01236 | Prolactin | 2.00 |
| P19827 | Inter-alpha-trypsin inhibitor heavy chain H1 | 2.04 |
| P12956 | X-ray repair cross-complementing protein 6 | 2.04 |
| P20700 | Lamin-B1 | 2.12 |
| P06702 | Protein S100-A9 | 2.22 |
| P07476 | Involucrin | 2.28 |
| P02746 | Complement C1q subcomponent subunit B | 2.38 |
| Q9NUQ9 | CYFIP-related Rac1 interactor B | 2.42 |
| P53634 | Dipeptidyl peptidase 1 | 2.48 |
| P46459 | Vesicle-fusing ATPase | 2.58 |
| P29992 | Guanine nucleotide-binding protein subunit alpha-11 | 2.73 |
| Q9UBC9 | Small proline-rich protein 3 | 2.74 |
| P47929 | Galectin-7 | 3.13 |
| P16144 | Integrin beta-4 | 3.20 |
| Q02487 | Desmocollin-2 | 3.22 |
| P29144 | Tripeptidyl-peptidase 2 | 3.42 |

and three being upregulated (S6 Fig, S2 Table). Eighty proteins were significantly changed for FGR with PreE versus FGR using peak intensity with twenty-eight being downregulated and fifty-two being upregulated. The largest changes were seen in Sideroflexin-3, Mannose-1-phosphate guanyltransferase alpha, Lectin, galactoside-binding, soluble, 7B and Desmoglein-3 (S7 Fig, Table 4). Zero proteins were significantly different for FGR with PreE versus FGR using spectral count analysis (S8 Fig).

## Discussion

The mechanisms of development of FGR and PreE are likely multifactorial and poorly understood. The umbilical cord may have either a direct role in the pathophysiology or may have changes secondary to the underlying process. Elucidating changes that are similar and different in the two conditions may help to uncover the underlying mechanism of the disease process and identify novel targets for therapeutics.

Wharton's Jelly is an important component of the umbilical cord and may contribute to the pathophysiology of many conditions including FGR and PreE. The normal development of the umbilical cord and Wharton's Jelly has been previously characterized [25, 26]. The umbilical cord and Wharton's Jelly increases in cross-sectional area until around 32 weeks' gestation at which point it levels off for the remainder of the pregnancy. Umbilical cords that have a larger cross-sectional area are primarily driven by increased vessel area. Umbilical cords that are smaller than average are due to decreased Wharton's Jelly area and are correlated to having a smaller placenta [26]. Previous studies have shown changes in the umbilical cord structure in FGR and PreE. The cross sectional area of the umbilical cord and Wharton's Jelly has been shown to be significantly smaller in FGR [9, 27]. Additionally, vascular changes have been seen in patients with PreE with umbilical artery tunica media areas larger in the outer layer, inner layer, and lumen [12].

ECM is a complex tissue that includes many proteins which form basement membranes and interstitial structures. The role of the ECM is to provide biochemical and structural support for surrounding tissues. Additionally, the ECM is essential for tissue hydration, storage of growth factors, and is involved in the inflammatory process [28]. ECM is typically composed of proteoglycans, non-proteoglycan polysaccharides (such as hyaluronic acid), and other proteins including collagen and elastin. Changes to the composition and ratio of these proteins impacts the tissue both in terms of biochemical function and mechanical stiffness. Because of the essential role of the ECM, disruption can lead to many diseases [29]. Disruption of ECM formation and composition may either be a causative component of changes seen in umbilical cord structure in FGR and PreE or as a result of the underlying mechanism of this disease.

Interestingly, there are intrinsic differences in the inflammatory response of the umbilical cord vasculature with significant changes in immune response factors in the umbilical artery and vein [30]. During infection, there is an increase in IL-1β and IL-8 mRNA in the umbilical vein. Fibrocytes arise from monocyte precursors and have features of tissue remodeling fibroblasts and macrophages [31]. Fetal fibroblasts migrate into the Wharton's Jelly via umbilical cord vasculature and there is significant decrease in the fibrocyte migration in FGR [30]. This decrease in fibrocytes in FGR may provide a mechanism for reduced stromal volume as fibroblasts are the major contributor to extracellular matrix production.

This study shows a significant decrease in umbilical cord area in FGR and FGR with PreE. This is driven by a decrease in the Wharton's Jelly area similar to previous studies [27]. There were no significant differences in the area of the umbilical artery and vein. Of note, the FGR with PreE group did have a higher mean area of the tunica media inner layer, but this did not

**Table 4. Intersection of significantly changed protein expression for FGR+PreE versus FGR peak intensity.**

| UniProt ID | Protein | Log Fold Change |
|---|---|---|
| Q9BWM7 | Sideroflexin-3 | -5.08 |
| Q96IJ6 | Mannose-1-phosphate guanyltransferase alpha; | -2.97 |
| P35998 | 26S proteasome regulatory subunit 7 | -2.97 |
| P08574 | Cytochrome c1, heme protein, mitochondrial | -2.87 |
| Q9UNS2 | COP9 signalosome complex subunit 3 | -2.69 |
| P30837 | Aldehyde dehydrogenase X, mitochondrial | -2.66 |
| P00367 | Glutamate dehydrogenase 1 | -2.54 |
| P07357 | Complement component C8 alpha chain | -2.41 |
| O15061 | Synemin; Type-VI intermediate filament | -2.25 |
| Q15717 | ELAV-like protein 1 | -2.19 |
| P35573 | Glycogen debranching enzyme | -2.17 |
| Q00534 | Cyclin-dependent kinase 6 | -2.15 |
| Q2M389 | WASH complex subunit 4 | -2.10 |
| O15230 | Laminin subunit alpha-5; | -2.05 |
| P54136 | Arginine—tRNA ligase, cytoplasmic | -2.02 |
| Q96QR8 | Transcriptional activator protein Pur-beta | -2.02 |
| Q9Y3A5 | Ribosome maturation protein SBDS | -2.01 |
| O95782 | AP-2 complex subunit alpha-1; | -1.99 |
| Q16891 | MICOS complex subunit MIC60 | -1.94 |
| Q08397 | Lysyl oxidase homolog 1; | -1.78 |
| P28066 | Proteasome subunit alpha type-5; | -1.64 |
| P08754 | Guanine nucleotide-binding protein G(k) subunit alpha | -1.49 |
| P22528 | Cornifin-B | -1.45 |
| P56134 | ATP synthase subunit f, mitochondrial | -1.41 |
| O43493 | Trans-Golgi network integral membrane protein 2 | -1.32 |
| P08648 | Integrin alpha-5; | -1.23 |
| P05556 | Integrin beta-1 | -1.19 |
| P12110 | Collagen alpha-2(VI) chain | -1.01 |
| Q07065 | Cytoskeleton-associated protein 4 | 0.78 |
| P30101 | Protein disulfide-isomerase A3 | 0.97 |
| Q13442 | 28 kDa heat- and acid-stable phosphoprotein | 1.21 |
| P08253 | type IV collagenase; | 1.30 |
| Q13347 | Eukaryotic translation initiation factor 3 subunit I; | 1.33 |
| Q14515 | SPARC-like protein 1 | 1.53 |
| Q01469 | Fatty acid-binding protein | 1.54 |
| O00339 | Matrilin-2 | 1.57 |
| Q96CX2 | BTB/POZ domain-containing protein KCTD12 | 1.58 |
| Q02809 | Procollagen-lysine,2-oxoglutarate 5-dioxygenase | 1.63 |
| O00391 | Sulfhydryl oxidase 1; | 1.70 |
| P25815 | Protein S100-P | 1.77 |
| P36952 | Serpin B5; | 1.88 |
| P24821 | Tenascin | 1.91 |
| Q16769 | Glutaminyl-peptide cyclotransferase | 1.93 |
| P55058 | Phospholipid transfer protein; | 2.05 |
| Q6UX71 | Plexin domain-containing protein 2 | 2.08 |
| O76061 | Stanniocalcin-2 | 2.11 |
| P52566 | Rho GDP-dissociation inhibitor 2 | 2.13 |

(*Continued*)

**Table 4.** (Continued)

| UniProt ID | Protein | Log Fold Change |
|---|---|---|
| P02771 | Alpha-fetoprotein; | 2.21 |
| Q9NRN5 | Olfactomedin-like protein 3 | 2.23 |
| P39059 | Collagen alpha-1(XV) chain; | 2.27 |
| P02746 | Complement C1q subcomponent subunit B | 2.29 |
| P17900 | Ganglioside GM2 activator | 2.35 |
| P06737 | Glycogen phosphorylase, liver form; | 2.38 |
| Q9BY89 | Uncharacterized protein KIAA167 | 2.44 |
| Q9Y446 | Plakophilin-3 | 2.44 |
| Q4ZHG4 | Fibronectin type III domain-containing protein 1 | 2.45 |
| Q9Y6R7 | Fc fragment of IgG binding protein | 2.49 |
| O94985 | Calsyntenin | 2.50 |
| Q08380 | Galectin-3-binding protein; | 2.53 |
| P10909 | Clusterin | 2.57 |
| P07476 | Involucrin | 2.65 |
| Q92817 | Envoplakin | 2.71 |
| Q15113 | Procollagen C-endopeptidase enhancer 1 | 2.73 |
| P12236 | ADP/ATP translocase 3 | 2.79 |
| P43251 | Biotinidase | 2.81 |
| P01861 | Immunoglobulin heavy constant gamma 4 | 2.81 |
| Q6P587 | Acylpyruvase FAHD1 | 2.89 |
| Q9UBC9 | Small proline-rich protein 3 | 2.93 |
| O95274 | Ly6/PLAUR domain-containing protein | 2.98 |
| P19320 | Vascular cell adhesion protein 1 | 3.01 |
| P12830 | Cadherin-1 | 3.05 |
| Q99542 | Matrix metalloproteinase-19 | 3.20 |
| Q8IVL6 | Prolyl 3-hydroxylase 3 | 3.35 |
| P16144 | Integrin beta-4; | 3.36 |
| Q9Y2B0 | Protein canopy homolog 2 | 3.41 |
| Q9Y240 | C-type lectin domain family 11 member A; | 3.47 |
| Q16787 | Laminin subunit alpha-3 | 3.50 |
| Q02487 | Desmocollin-2 | 3.69 |
| P32926 | Desmoglein-3 | 4.39 |
| P47929 | Lectin, galactoside-binding, soluble, 7B; | 5.43 |

reach significance due to wide variability. The larger amount of variability in the FGR with PreE group may be driven by a separate mechanism underlying the disease process in PreE.

The data from this study shows significant changes across many cellular components and functions. The majority of proteins identified as downregulated in the FGR with PreE versus control group are involved with the extracellular matrix, RNA damage and repair, immune response, cell death pathways and cellular function. Proteins that were upregulated are involved with oxidative stress response, immune response, and cell adhesion. We also found a significant downregulation of pregnancy-zone protein in FGR with PreE versus controls.

When comparing FGR versus controls, the proteins identified fall into very different cellular function pathways. The majority of proteins that were downregulated in the FGR groups are involved with extracellular matrix formation including procollagen, collagen alpha-1 chain, and matrillin. Pregnancy-zone protein was also found to be downregulated in the FGR

group as was alpha-fetoprotein. The physiologic role of pregnancy-zone protein and alpha-fetoprotein in the umbilical cord is unknown. The majority of proteins that were upregulated were involved with transcription factors, cell adhesion, ECM organization, and golgi transport.

The comparison of FGR with PreE versus isolated FGR revealed eighty significantly changed proteins. Inflammatory response proteins had mixed expression changes across the different groups. Many of the downregulated proteins are involved with cell energy pathways, exocytosis, and metabolic pathways. Upregulated pathways include vasculature remodeling, angiogenesis, and ECM composition and development. Previous studies have shown an upregulation of pro-inflammatory substances and mixed changes in angiogenic and anti-angiogenic factors in isolated PreE [32]. Here we see a similar mixture of up and down-regulation of angiogenic factors likely displaying the complex nature of the PreE process.

This study is novel as it is the first to use proteomics to specifically look at the proteomic changes in the Wharton's Jelly, which has been understudied compared to other tissue such as maternal and fetal blood and placenta. The results of this study show the significant changes in the proteins that serve as significant components of the ECM and likely contribute to the significant reductions in umbilical cord cross-sectional area and Wharton's Jelly. A strength of this study is the novel use of five protein search and inference engines for identification of proteins from the mass spectrometry data as well as using both spectral count and peak intensity methods. By using the overlap data of commonly identified proteins that were significantly changed, this increases the confidence that this is a true change.

A recent meta-analysis of proteomic biomarkers for PreE identified many differentially expressed proteins in multiple biological samples including maternal blood, placenta, umbilical cord blood and urine [14]. Our study demonstrated multiple significant changes in proteins highlighted in this meta-analysis. Fibrinogen alpha chain was found to be increased in the PreE+FGR group compared the FGR. This protein was also found to be increased in blood and urine samples in the meta-analysis. Similar patterns were seen for clusterin. We found that pregnancy -zone protein was significantly reduced in our FGR versus control group. The meta-analysis showed a significant decrease in this protein in blood, but an increase in plasma and urine in women with PreE. The meta-analysis study highlights the complex differential expression of proteins in different tissue samples from women with PreE. Our study adds to the knowledge base by including an additional tissue type further demonstrating the complexity of protein expression between both FGR and FGR+PreE.

Limitations to the study include a small number of patients for each group. Increasing the numbers may help to decrease the variance in the histological data and proteomic data. The route of delivery was not proportional in each of the groups, but route of delivery is unlikely to contribute to change in protein expression or structure. Another limitation is that there was no isolated PreE group for comparison. It is interesting to see the proteomic changes with those with FGR that did develop PreE compared to isolated FGR as they are likely distinct process, but it does not allow for comparison of isolated PreE versus controls which all the other studies have previously compared. Additionally, the proteomics results can serve as a screening tool where proteins identified as significantly changed will need to be confirmed by traditional validation methods such as western blotting.

Identification of changes to the proteomic profiles and mechanism of umbilical cord morphology changes may identify novel targets as potential mechanisms for FGR and PreE. We demonstrated that there is a significant reduction in umbilical cord area and Wharton's Jelly area in FGR, similar to previous studies. Proteomic analysis showed changes in extracellular matrix, cellular process, inflammatory pathway, and angiogenesis proteins across the group comparisons. These results display the complex nature of the changes in the umbilical cord

and additional research into these changes may help to identify the mechanisms behind FGR and PreE. Further work in validating these proteomic differences may enable identification of novel molecules or pathways for therapeutic targets for these conditions.

## Supporting information

**S1 Fig. Artery average tunica media outer layer area.** There were no significant differences (p = 0.466).
(TIF)

**S2 Fig. Artery average tunica media inner layer area.** There were no significant differences (p = 0.754).
(TIF)

**S3 Fig. Umbilical cord vein area.** There were no significant differences (p = 0.794).
(TIF)

**S4 Fig. Overlap of significantly changed protein ID's from mass spectrometry search and inference engines for spectral count of FGR versus controls.** Legend (CF: comet fido; CE: comet epifany; XE: X!Tandem epifany; MF: MSGF fido; ME: MSGF epiphany).
(TIF)

**S5 Fig. Overlap of significantly changed protein ID's from mass spectrometry search and inference engines for peak intensity of FGR+PreE versus controls.** Legend (CF: comet fido; CE: comet epifany; XE: X!Tandem epifany; MF: MSGF fido; ME: MSGF epiphany).
(TIF)

**S6 Fig. Overlap of significantly changed protein ID's from mass spectrometry search and inference engines for spectral count of FGR+PreE versus controls.** Legend (CF: comet fido; CE: comet epifany; XE: X!Tandem epifany; MF: MSGF fido; ME: MSGF epiphany).
(TIF)

**S7 Fig. Overlap of significantly changed protein ID's from mass spectrometry search and inference engines for peak intensity of FGR+PreE versus FGR.** Legend (CF: comet fido; CE: comet epifany; XE: X!Tandem epifany; MF: MSGF fido; ME: MSGF epiphany).
(TIF)

**S8 Fig. Overlap of significantly changed protein ID's from mass spectrometry search and inference engines for spectral count of FGR+PreE versus FGR.** Legend (CF: comet fido; CE: comet epifany; XE: X!Tandem epifany; MF: MSGF fido; ME: MSGF epiphany).
(TIF)

**S1 Table. Intersection of significantly changed protein expression for FGR versus controls peak intensity.**
(DOCX)

**S2 Table. Intersection of significantly changed protein expression for FGR+PreE versus controls spectral count.**
(DOCX)

## Acknowledgments

We would like to thank the CCIC Proteomics Core with help from Liwen Zhang, PhD, Sophie Harvey, PhD.

## Author Contributions

**Conceptualization:** Miranda L. Gardner, Michael A. Freitas, Kara M. Rood, Marwan Ma'ayeh.

**Data curation:** Miranda L. Gardner.

**Formal analysis:** Matthew S. Conrad, Miranda L. Gardner, Christine Miguel, Kara M. Rood, Marwan Ma'ayeh.

**Investigation:** Matthew S. Conrad.

**Methodology:** Matthew S. Conrad, Michael A. Freitas, Kara M. Rood, Marwan Ma'ayeh.

**Supervision:** Michael A. Freitas, Kara M. Rood.

**Writing – original draft:** Matthew S. Conrad, Miranda L. Gardner, Kara M. Rood, Marwan Ma'ayeh.

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
