## [Decision Letter · Decision Letter 0]

23 Nov 2021

PONE-D-21-32980Proteomic analysis of the umbilical cord in fetal growth restriction and preeclampsiaPLOS ONE

Dear Dr. Rood,

Thank you for submitting your manuscript to PLOS ONE. After careful consideration, we feel that it has merit but does not fully meet PLOS ONE’s publication criteria as it currently stands. Therefore, we invite you to submit a revised version of the manuscript that addresses the points raised during the review process.

We look forward to receiving your revised manuscript.

Kind regards,

Robert A. Niederman, Ph.D

Academic Editor

PLOS ONE

Journal Requirements:

"This work was supported by by the CCTS and CTSA Grant number (UL1TR002733).  We would like to thank the CCIC Proteomics Core with help from Liwen Zhang, PhD, Sophie Harvey, PhD."

"This work was supported by by the CCTS and CTSA Grant number (UL1TR002733) to MSC.

https://ccts.osu.edu/content/current-grant-opportunities

The tims-ToF Pro was purchased using funds from NIH award S10 OD026945.

The funders had no role in study design, data collection and analysis, decision to publish, or preparation of the manuscript"

Additional Editor Comments:

Editors comments:

This is an important Ms. insofar as it demonstrates that in fetal growth restricted pregnancies and in preeclampsia, the proteome of the Wharton’s Jelly component of the umbilical cord is largely altered when compared to that of a normal control group. The affected proteins included those connected with extracellular matrix, cellular processes, inflammation and angiogenesis, and it is suggested that these data could ultimately contribute to an improved understanding of the etiology of fetal growth restriction and preeclampsia.

The reviewers have provided detailed suggestions for necessary improvements of the Ms. and these must be satisfactorily dealt with by the authors before this Ms. can be reconsidered for publication.

Some additional changes suggested for the text:

Should use min, sec, h throughout

Table 1: Maternal Age (years) FGR value incorrect? 31.6 (50)? Should this be: 31.6 (5.0)??

Fig. 1: Increase Y-axis legend font size as in Suppl. Fig.1: Area (mm2)??

Suppl. Fig. 1: IUGR??

P. 23, ¶2, line 2: immune response factors

P. 24, ¶3, line 5: the majority of proteins

Reviewers' comments:

Reviewer's Responses to Questions

**Comments to the Author**

1. Is the manuscript technically sound, and do the data support the conclusions?

Reviewer #1: Yes

Reviewer #2: Yes

2. Has the statistical analysis been performed appropriately and rigorously? 

Reviewer #1: Yes

Reviewer #2: Yes

3. Have the authors made all data underlying the findings in their manuscript fully available?

Reviewer #1: Yes

Reviewer #2: Yes

4. Is the manuscript presented in an intelligible fashion and written in standard English?

Reviewer #1: Yes

Reviewer #2: Yes

5. Review Comments to the Author

Reviewer #1: The work presented by Conrad et al. examined changes in the umbilical cord (UC) composition in pregnancies complicated by FGR and FGR with PreE by histologic and proteomic analysis for Wharton’s Jelly samples.

Introduction reads well and the study design is technically valid. But

My comments to the author are listed below:

• The first word in the title is incomplete (I think the author means Proteomic)

• In the last step of sample preparation for proteomics, why the supernatant was not desalted first before drying on vacufuge instead of drying then desalting and drying again.

• In Histological analysis of the umbilical cord and Wharton’s Jelly area please specify which groups were significantly different according to the post-hok results of the ANOVA. This can also be indicated in the legend of Figure 1. Please Add the statistical test to the legend of figure 1.

• The flow of the proteomics section in the results is difficult. The authors only summarize the results rather than describing it. A table that summarizes number of changed proteins using spectral count and peak intensity methods between different compared group might be added.

• According to the author “A strength of this study is the novel use of five protein search and inference engines for identification of proteins from the mass spectrometry data as well as using both spectral count and peak intensity methods. By using the overlap data of commonly identified proteins that were significantly changed, this increases the confidence that this is a true change”.

Any common proteins that were significantly changed using spectral count and peak intensity? Why authors used two methods if they end with different altered proteins for the same set of samples when using the two analytical methods? Were proteins significantly altered by the two methods involved in similar biochemical processes?

• In the discussion of the proteomics findings, the author only mentioned the pathways in which the proteins are involved without linking this to underlying mechanisms involved in preeclampsia or FGR. Please highlight how these proteins are related to the pathophysiology of the studied disorders

• How proteomic changes in the Wharton’s Jelly can be linked to other proteomic studies that investigated different sample types including placenta and blood.

• the small sample size (n=5) in each group is of concern and the authors mentioned this in the limitation section.

• The authors made 6 binary comparisons (using two analytical methods). I was wondering how they decided for specific comparisons to be presented in the text while others as supplementary

• Please be consistent (e.g., either PreE or preeclampsia)

Reviewer #2: Re: Proteomic analysis of the umbilical cord in fetal growth restriction and preeclampsia

Manuscript Number: PONE-D-21-32980

This is a proteomic analysis of the umbilical cord in a case control study of FGR, FGR with preeclampsia and control pregnancies. It is novel, well-organized and well-written.

Comments:

1. Introduction, fifth paragraph. The patients are described as “pre-eclamptic women.” Respectfully, they should not be defined by their disease and should be referred to as “women with preeclampsia.”

2. Introduction, last sentence. “Specifically, these changes will include decreased cross-sectional area of the umbilical cord and proteomic changes in extracellular matrix protein composition that contribute to formation of the Wharton’s Jelly.” The phrase “that contribute to the formation of Wharton’s Jelly” suggests that some property of Wharton’s Jelly will be enhanced (and is kind of non-specific). I do not believe the data support this. If so, one of two responses by the authors is appropriate. Either rephrase this hypothesis to suggest that nothing is “enhanced” or include a discussion (in the discussion section) as to why this part of the hypothesis was not confirmed.

3. Methods. Parity is not seen in the available data. Parity has a tremendous effect on many outcomes of pregnancy. If there are any data in the literature showing that parity has no effect on umbilical cord volume or other parameters, parity could be sefaly omitted. In the absence of any such data, parity should be included in the demographic data.

4. Methods, Sample preparation for proteomics. In this paragraph, the biologic tissue is separated into supernatant and pellet. The authors use the word “sample” numerous times. For clarity, I believe it is best to specify supernatant or pellet each time the word "sample" appears.

5. Methods, LC-MS/MS. This paragraph starts with, “Protein identification was performed . . .” For clarity, I believe is is best to specify the sample used (Supernatant or pellet? Wharton’s Jelly or vessel?)

6. Results, Histology. Regarding the use of the phrase “cesarean section,” a more respectful and contemporary phrase is “cesarean delivery.” “Cesarean section” only refers to the act of cutting, not the delivery.

7. Table 1. Apgars not normally distributed. Should properly be shown as median (range or IQR).

8. Figures. For clarity, whenever “PreE” appears, it should be “FGR with PreE.”

9. Figures. Figures should stand alone, thus, abbreviations should be explained in footnotes. (FGR, PreE, Cont)

10. Supplemental Figure 2. “IUGR” should read “FGR.”

6. PLOS authors have the option to publish the peer review history of their article (what does this mean?). If published, this will include your full peer review and any attached files.

Reviewer #1: No

Reviewer #2: **Yes: **Daniel W Skupski, MD

---

## [Author Response · Author response to Decision Letter 0]

13 Dec 2021

Response to Reviewers:

Please see our response to the editor and reviewer comments below. Our responses are indicated in bold.

Additional Editor Comments:

Editors comments:

This is an important Ms. insofar as it demonstrates that in fetal growth restricted pregnancies and in preeclampsia, the proteome of the Wharton’s Jelly component of the umbilical cord is largely altered when compared to that of a normal control group. The affected proteins included those connected with extracellular matrix, cellular processes, inflammation and angiogenesis, and it is suggested that these data could ultimately contribute to an improved understanding of the etiology of fetal growth restriction and preeclampsia.

The reviewers have provided detailed suggestions for necessary improvements of the Ms. and these must be satisfactorily dealt with by the authors before this Ms. can be reconsidered for publication.

Some additional changes suggested for the text:

Should use min, sec, h throughout – This has been changed to be consistent 

Table 1: Maternal Age (years) FGR value incorrect? 31.6 (50)? Should this be: 31.6 (5.0)?? – yes, this has been corrected

Fig. 1: Increase Y-axis legend font size as in Suppl. Fig.1: Area (mm2)??

Suppl. Fig. 1: IUGR?? – this has been corrected

P. 23, ¶2, line 2: immune response factors – this has been changed

P. 24, ¶3, line 5: the majority of proteins– this has been changed

Reviewers' comments:

Reviewer's Responses to Questions

 

Comments to the Author

5. Review Comments to the Author

Reviewer #1: The work presented by Conrad et al. examined changes in the umbilical cord (UC) composition in pregnancies complicated by FGR and FGR with PreE by histologic and proteomic analysis for Wharton’s Jelly samples.

Introduction reads well and the study design is technically valid. But

My comments to the author are listed below:

• The first word in the title is incomplete (I think the author means Proteomic) – this has been corrected

• In the last step of sample preparation for proteomics, why the supernatant was not desalted first before drying on vacufuge instead of drying then desalting and drying again. – this is the standard protocol that our proteomics core facility uses for preparing the samples

• In Histological analysis of the umbilical cord and Wharton’s Jelly area please specify which groups were significantly different according to the post-hok results of the ANOVA. This can also be indicated in the legend of Figure 1. Please Add the statistical test to the legend of figure 1. -this information has been added to the methods, results section, and to the legend of figure 1.

• The flow of the proteomics section in the results is difficult. The authors only summarize the results rather than describing it. A table that summarizes number of changed proteins using spectral count and peak intensity methods between different compared group might be added. – we chose to summarize the results with number of differences with highlights of significantly changed proteins due to the abundance of data. The number of changed proteins in the overlap data are found in the figures. We did not want to duplicate this information by adding another table.

• According to the author “A strength of this study is the novel use of five protein search and inference engines for identification of proteins from the mass spectrometry data as well as using both spectral count and peak intensity methods. By using the overlap data of commonly identified proteins that were significantly changed, this increases the confidence that this is a true change”.

Any common proteins that were significantly changed using spectral count and peak intensity? Why authors used two methods if they end with different altered proteins for the same set of samples when using the two analytical methods? Were proteins significantly altered by the two methods involved in similar biochemical processes? – there are many analysis methods for proteomic data including both detection (spectral count vs peak intensity) and protein identification (protein search and inference engines). Unfortunately, there is no consensus about the best method for detection and analysis. By using both the spectral count and peak intensity, we are providing all of the possible changes which are identified rather than restricting it to just one method which may not detect all of the changes. We chose to use the intersection data between the five inference engines to help improve the confidence that the changes found are a true change.

• In the discussion of the proteomics findings, the author only mentioned the pathways in which the proteins are involved without linking this to underlying mechanisms involved in preeclampsia or FGR. Please highlight how these proteins are related to the pathophysiology of the studied disorders – many of the proteins found to be different have a role in extracellular matrix which directly contributes to the umbilical cord structure. Additional description of this has been added to the discussion. A large majority of the proteins identified can be grouped into overall “functional classification”, but the specific role of these proteins in both the umbilical cord and in these disease processes are unknown. Further research will hope to further characterize the physiologic role of these proteins.

• How proteomic changes in the Wharton’s Jelly can be linked to other proteomic studies that investigated different sample types including placenta and blood. – we have modified the paragraph in the discussion to include more direct comparison of our findings with a recent meta-analysis of proteomic biomarkers in pre-eclampsia.

• the small sample size (n=5) in each group is of concern and the authors mentioned this in the limitation section. – Yes, the sample size is small for each group, but we did find significant differences even with this sample size. We hope future studies will expand on this work with a higher sample size.

• The authors made 6 binary comparisons (using two analytical methods). I was wondering how they decided for specific comparisons to be presented in the text while others as supplementary – We chose the tables included in the main text as they highlighted major comparison differences between the three group comparisons.

• Please be consistent (e.g., either PreE or preeclampsia) – this has been corrected

 

Reviewer #2: Re: Proteomic analysis of the umbilical cord in fetal growth restriction and preeclampsia

Manuscript Number: PONE-D-21-32980

This is a proteomic analysis of the umbilical cord in a case control study of FGR, FGR with preeclampsia and control pregnancies. It is novel, well-organized and well-written.

Comments:

1. Introduction, fifth paragraph. The patients are described as “pre-eclamptic women.” Respectfully, they should not be defined by their disease and should be referred to as “women with preeclampsia.” – this has been corrected

2. Introduction, last sentence. “Specifically, these changes will include decreased cross-sectional area of the umbilical cord and proteomic changes in extracellular matrix protein composition that contribute to formation of the Wharton’s Jelly.” The phrase “that contribute to the formation of Wharton’s Jelly” suggests that some property of Wharton’s Jelly will be enhanced (and is kind of non-specific). I do not believe the data support this. If so, one of two responses by the authors is appropriate. Either rephrase this hypothesis to suggest that nothing is “enhanced” or include a discussion (in the discussion section) as to why this part of the hypothesis was not confirmed. – this has been modified and “contribute to the formation of the Wharton’s Jelly” was removed.

3. Methods. Parity is not seen in the available data. Parity has a tremendous effect on many outcomes of pregnancy. If there are any data in the literature showing that parity has no effect on umbilical cord volume or other parameters, parity could be sefaly omitted. In the absence of any such data, parity should be included in the demographic data. – parity data has been added in the methods, results, and table 1. There was no significant difference in parity between the groups.

4. Methods, Sample preparation for proteomics. In this paragraph, the biologic tissue is separated into supernatant and pellet. The authors use the word “sample” numerous times. For clarity, I believe it is best to specify supernatant or pellet each time the word "sample" appears. – the wording has been clarified with identification of the supernatant and pellet layers in the steps that had a pellet formation.

5. Methods, LC-MS/MS. This paragraph starts with, “Protein identification was performed . . .” For clarity, I believe is is best to specify the sample used (Supernatant or pellet? Wharton’s Jelly or vessel?) - this has been clarified in the methods

6. Results, Histology. Regarding the use of the phrase “cesarean section,” a more respectful and contemporary phrase is “cesarean delivery.” “Cesarean section” only refers to the act of cutting, not the delivery. – this has been corrected

7. Table 1. Apgars not normally distributed. Should properly be shown as median (range or IQR). -Table one has been modified with APGARs reported as medians with range

8. Figures. For clarity, whenever “PreE” appears, it should be “FGR with PreE.” – this has been corrected

9. Figures. Figures should stand alone, thus, abbreviations should be explained in footnotes. (FGR, PreE, Cont) – an abbreviation legend was added to figure 1

10. Supplemental Figure 2. “IUGR” should read “FGR.”

6. PLOS authors have the option to publish the peer review history of their article (what does this mean?). If published, this will include your full peer review and any attached files.

Do you want your identity to be public for this peer review? For information about this choice, including consent withdrawal, please see our Privacy Policy.

Reviewer #1: No

Reviewer #2: Yes: Daniel W Skupski, MD

---

## [Editor Report · Decision Letter 1]

16 Dec 2021

Proteomic analysis of the umbilical cord in fetal growth restriction and preeclampsia

PONE-D-21-32980R1

Dear Dr. Rood,

We’re pleased to inform you that your manuscript has been judged scientifically suitable for publication and will be formally accepted for publication once it meets all outstanding technical requirements.

Kind regards,

Robert A. Niederman, Ph.D

Academic Editor

PLOS ONE
---

## [Editor Report · Acceptance letter]

30 Dec 2021

PONE-D-21-32980R1 

Proteomic analysis of the umbilical cord in fetal growth restriction and preeclampsia

Dear Dr. Rood:

I'm pleased to inform you that your manuscript has been deemed suitable for publication in PLOS ONE. Congratulations! Your manuscript is now with our production department. 

Kind regards, 

on behalf of

Dr. Robert A. Niederman 

Academic Editor

PLOS ONE